# The experience of European hospital-based health care workers on following infection prevention and control procedures and their wellbeing during the first wave of the COVID-19 pandemic

**Denise van Hout**[1¤]*, **Paul Hutchinson**[2], **Marta Wanat**[3], **Caitlin Pilbeam**[3], **Herman Goossens**[4], **Sibyl Anthierens**[5], **Sarah Tonkin-Crine**[3,6], **Nina Gobat**[3]

**1** Julius Center for Health Sciences and Primary Care, University Medical Center Utrecht, University Utrecht, Utrecht, The Netherlands, **2** Department of Global Community Health and Behavioral Sciences, School of Public Health and Tropical Medicine, Tulane University, New Orleans, LA, United States of America, **3** Nuffield Department of Primary Care Health Sciences, University of Oxford, Oxford, United Kingdom, **4** Laboratory of Medical Microbiology, Vaccine and Infectious Disease Institute, University of Antwerp, Antwerp, Belgium, **5** Department of Family Medicine and Population Health (FAMPOP), University of Antwerp, Antwerp, Belgium, **6** National Institute for Health Research Health Protection Research Unit in Healthcare Associated Infections and Antimicrobial Resistance, University of Oxford, Oxford, United Kingdom

¤ Current address: Centre for Infectious Diseases, Epidemiology and Surveillance, National Institute for Public Health and the Environment, Bilthoven, The Netherlands

* denise.van.hout@rivm.nl

## Abstract

### Background

Working under pandemic conditions exposes health care workers (HCWs) to infection risk and psychological strain. A better understanding of HCWs' experiences of following local infection prevention and control (IPC) procedures during COVID-19 is urgently needed to inform strategies for protecting the psychical and psychological health of HCWs. The objective of this study was therefore to capture the perceptions of hospital HCWs on local IPC procedures and the impact on their emotional wellbeing during the first wave of the COVID-19 pandemic in Europe.

### Methods

Participants were recruited in two sampling rounds of an international cross-sectional survey. Sampling took place between 31 March and 17 April 2020 via existing research networks and between 14 May and 31 August 2020 via online convenience sampling. Main outcome measures were behavioural determinants of HCWs' adherence to IPC guidelines and the WHO-5 Well-Being Index, a validated scale of 0–100 reflecting emotional wellbeing. The WHO-5 was interpreted as a score below or above 50 points, a cut-off score used in previous literature to screen for depression.

**Data Availability Statement:** A generic version of the protocol and survey tool of this study are freely available via the Research & Development COVID-

19 Blueprint of the World Health Organization (WHO) to be used in different clinical settings globally. All relevant data are within the manuscript and its supporting information files. Data are collected as part of the RECOVER Consortium. A public repository of all data collected within RECOVER is still under construction. Until that is available, researchers who are interested in raw data of the current paper are welcomed to contact the RECOVER Consortium through Frank Leus (Head of Data Management at RECOVER), at F.R. Leus@umcutrecht.nl.

**Funding:** This manuscript is part of the output from RECOVER (Rapid European COVID-19 Emergency research Response), which has received funding from the EU Horizon 2020 research and innovation programme (grant agreement number 101003589). The funder had no role in the writing of the manuscript or the decision to submit for publication. STC received additional funding from the NIHR Health Protection Research Unit in Healthcare Associated Infections and Antimicrobial Resistance at the University of Oxford in partnership with Public Health England (PHE). The views expressed are those of the authors and not necessarily those of the National Health Service, the NIHR, the Department of Health and Social Care, or PHE.

**Competing interests:** The authors have declared that no competing interests exist.

## Results

2289 HCWs from 40 countries in Europe participated. Mean age was 42 (±11) years, 66% were female, 47% and 39% were medical doctors and nurses, respectively. 74% (n = 1699) of HCWs were directly treating patients with COVID-19, of which 32% (n = 527) reported they were fearful of caring for these patients. HCWs reported high levels of concern about COVID-19 infection risk to themselves (71%) and their family (82%) as a result of their job. 40% of HCWs considered that getting infected with COVID-19 was not within their control. This feeling was more common among junior than senior HCWs (46% versus 38%, *P* value < .01). Sufficient COVID-19-specific IPC training, confidence in PPE use and institutional trust were positively associated with the feeling that becoming infected with COVID-19 was within their control. Female HCWs were more likely than males to report a WHO-5 score below 50 points (aOR 1.5 (95% confidence interval (CI) 1.2–1.8).

## Conclusions

In Europe, the COVID-19 pandemic has had a differential impact on those providing direct COVID-19 patient care, junior staff and women. Health facilities must be aware of these differential impacts, build trust and provide tailored support for this vital workforce during the current COVID-19 pandemic.

## Introduction

Health care workers (HCWs) provide direct treatment and care for patients with coronavirus disease 2019 (COVID-19), as well as for those requiring ongoing care not related to COVID-19. This role exposes them to occupational hazards that may impact their health and wellbeing, including increased exposure to severe acute respiratory syndrome coronavirus-2 (SARS-CoV-2), burnout, and stigma [1]. Protecting the physical and psychological health of HCWs is a key priority. Not only is there a moral obligation to support those willing to provide medical care in a public health emergency, this support is also vital for ensuring sufficient levels of skilled staff and continued functioning of health facilities during and after the COVID-19 pandemic [2]. Importantly, the exact need for support is heterogeneous and could vary depending on aspects such as healthcare institution location, or type, but also on HCWs' personal characteristics, such as seniority level, job role or gender. Studies published in the first months of the COVID-19 pandemic highlighted increased signs of depression, anxiety, insomnia and distress in HCWs [3–8], with a differential impact on women [3,9]. United Nations Women stressed the importance of attention to potential gender differences when researching the impact of the COVID-19 pandemic, as this information is needed to inform policy responses [10].

While HCWs can acquire COVID-19 through many different routes, including via infected family members or through general community transmission [11], their work in health facilities clearly places them at increased risk of exposure to SARS-CoV-2 [2,12–14]. Standardized and robust infection prevention control (IPC) procedures are the primary approach to reducing transmission in health facilities. Effective implementation of IPC procedures requires engagement and commitment from those administering health facilities. This includes ensuring that local policies are developed and made available; systems and processes are set up for recommended procedures; sufficient and ongoing supply of materials are accessible; staff have

access to training and support materials; and processes for monitoring staff-level adherence to local policies are in place.

While one of the main goals of IPC guidelines is the prevention of nosocomial transmission, another key aspect is that they should ensure that HCWs feel they can protect themselves from getting infected. Research conducted during the SARS epidemic in 2003 demonstrated the importance of organizational and social factors and identifed how feeling prepared and confident in their ability to deliver effective IPC was critical to protect both HCWs' physical and psychological health [15]. At the onset of the COVID-19 pandemic, large-scale information about the perceptions of European hospital HCWs from multiple countries across Europe was not yet available. Understanding HCWs' experiences of following locally recommended IPC procedures during COVID-19 were highly needed to inform strategies for better engaging and supporting HCWs to protect themselves and maintain their wellbeing.

The objective of this study was therefore to rapidly capture hospital HCWs' perceptions on local IPC procedures and the impact on their emotional wellbeing during the first wave of the COVID-19 pandemic in Europe, and to explore potential differences among hospital HCWs with different seniority level, job role and gender.

## Methods

### Study design

We performed two sampling rounds of a cross-sectional survey among hospital HCWs in European countries during the peak of the first wave of the COVID-19 pandemic in Europe (31 March 2020–17 April 2020) and during the aftermath of the first wave (14 May 2020–31 August 2020).

### Sampling and recruitment

We invited European HCWs providing medical care in hospital settings. The first sampling round was conducted during the peak pandemic wave in Europe. Because a low response rate was observed–in part due to competing clinical pressures during this time—a second round of data collection was performed in which we adapted our approach to recruitment and performed online convenience sampling. For the first sampling round, we recruited participants through two European hospital research networks: Combatting Bacterial Resistance in Europe (COMBACTE) and the Spanish Biomedical Research Networking Center (CIBER). For the second sampling round, we recruited participants via newsletters in research networks, clinical networks and social media channels by distributing an online study flyer.

### Survey tool

A data collection tool was rapidly developed for a generic World Health Organization (WHO) protocol to meet the aims of the study (see S1 File) [16]. Experts in the Social Science and IPC Working Group, under the COVID-19 Research Roadmap, identified a pool of items based on WHO IPC interim guidance published in March 2020 [17]. These items were developed to capture theoretically informed influences on HCWs' motivation, opportunity, and ability to follow general IPC precautions. Items were designed for responses on a 7-point Likert scale, ranging from 'Strongly disagree' to 'Strongly agree'. Once identified, the Theoretical Domains Framework (TDF), previously used for studying clinician behaviour, was used to inform the selection of the items [18–20]. Based on a previously validated measure, HCWs' trust in the institution where they worked was assessed by capturing three dimensions of institutional trust (competence, honesty, act in best interests of staff) [21]. Finally, the WHO-5 Well-Being

Index (WHO-5) was included, a validated short and generic global rating scale measuring subjective wellbeing during the last two weeks [22]. Information was also collected on participant demographics, including experience of caring for patients with suspected or confirmed COVID-19 infection. Due to the time constraints under which this tool was developed, pretesting, reliability tests, and validation could not be conducted prior to data collection. As a result, data collected in the first sampling round of this study were used to investigate the psychometric properties of the survey tool and refine the items for data collection in the second sampling round.

## Electronic data collection

The survey was only available in English. The electronic data capturing (EDC) systems Castor v2020.1.9 and Qualtrics survey platform (Provo, Utah) were used for data collection in the first and second sampling round, respectively.

## Statistical analyses

As participation was anonymous, we could not prevent HCWs participating in both survey rounds. Therefore, respondents from round 2 with identical demographic information as HCWs from round 1 (age, gender, country and specialism) were excluded from further analyses (n = 16 exclusions).

Descriptive statistics were used to describe HCWs' perceptions of IPC procedures. Absolute numbers of respondents were provided alongside percentages, as denominators differed slightly per survey item due to respondent in-survey drop out. To quantify HCWs' emotional wellbeing, responses to the five WHO-5 statements were summarised into a total raw score and multiplied by 4 to produce an individual total score from 0 to 100, with the higher end of the scale representing best possible wellbeing [23]. Interpretation of the WHO-5 Well-Being score considers that a cut-off score of <50 is used when screening for clinical depression [23]. Differences in WHO-5 scores were assessed for gender, job role, European region and providing COVID-19 care by independent sample T-test or one-way ANOVA, depending on the number of groups compared. To assess differences in wellbeing between male and female HCWs we estimated the independent effect of gender on a WHO-5 Well-Being Index below 50 points using logistic regression, including predefined control variables for age, living situation (i.e., living alone or sharing a household), European region, job role, hospital type and providing COVID-19 patient care.

Multivariable regression was performed to examine the association between prespecified behavioral determinants and HCWs' perceived sense of control over getting infected with COVID-19. The association between the feeling of control over getting infected and a WHO-5 Well-Being score <50 was investigated using the same model as for the effect of gender, adding the sense of control statement. From the Likert-scale questions, we conducted principal component analysis to examine correlation matrices and construct indices from the first principal component for the following measures: beliefs about effectiveness of PPE, availability of PPE at the respondent's institution, and skills for preparedness for dealing with COVID-19 (see S2 File). The index constructed from the first principal component for each of these measures was then included as an explanatory variable in the multivariable analysis. This allowed us to test, for example, for the independent effect of availability of PPE at the respondent's institution, on the dependent variable, HCWs' perceived sense of control over getting infected with COVID-19, while controlling for other regression model covariables. For each of the constructed indices, Cronbach's alpha scores for the Likert-scale components exceeded 0.75.

For all analyses, a two-tailed *P* value < .05 was considered statistically significant. There was no formal sample size calculation before the start of this study. Surveys with completion of only demographic and basic IPC training information (corresponding to survey completion of <58%) were excluded. All analyses were performed with Statistical Package for Social Sciences V.25.0.2 (SPSS, Chicago, Illinois, USA) and R Version 3.4.1.

### Ethics

This study was conducted in accordance with the EU GDPR (General Data Protection Regulation) and the principles of the Declaration of Helsinki [24]. The Medical Research Ethics Committee of the University Medical Center (UMC) Utrecht waived the need for extensive ethical review (IRB correspondence number 18-574C). Electronic individual informed consent for participation was obtained at the start of the survey.

## Results

### Study population

In total, 2289 hospital HCWs participated (round 1: n = 190, round 2: n = 2099). HCWs worked in 40 European countries, the majority in Southern Europe (n = 1244, 54%) (S1 Table). Forty-eight percent (n = 1088) had experience working in a clinical setting during a previous epidemic with a novel respiratory virus such as SARS, MERS-CoV or H1N1. Sixty-eight percent (n = 129) and 75% (n = 1570) had personally cared for a patient with suspected or confirmed COVID-19 in the first and second sampling round, respectively. Detailed demographic information is provided in Table 1.

### Wellbeing

There were 2180 (95%) HCWs who completed all questions about emotional wellbeing. Overall mean WHO-5 Well-Being Index was 56.3 (±19.3) and was slightly higher among HCWs that participated in the second sampling round than in the first round (56.7 ±19.0 versus 51.9 ±22.0, respectively). Scores differed per region, being lowest in Eastern Europe (52.7 ±19.9) and highest in Western Europe (62.1 ±17.8) (*P* value < .001). Junior nurses and medical doctors had lower scores compared to their senior counterparts (*P* value < .05) (Table 2). Overall prevalence of a WHO-5 Well-Being Index below 50 points was 38% (95% confidence interval (CI) 36%-40%). Mean WHO-5 was 59.6 ±19.3 for male and 54.5 ±19.0 for female HCWs (*P* value < .001). In multivariable logistic regression, female HCWs had higher risk of a WHO-5 score below 50 points compared to males (aOR 1.5 (95% confidence interval (CI) 1.2–1.8) (S3 Table).

A large proportion of HCWs reported concerns about the risk for themselves of becoming ill (1568, 71%) and about the risk to their family related to COVID-19 as a result of their job role (1809, 82%) (Fig 1). Thirty-two percent (n = 527) of HCWs directly caring for patients with COVID-19 reported being afraid of looking after these patients. Significant additional strain to their workload due to following recommended IPC procedures was reported by 86% (n = 1463) of these HCWs. HCWs that participated in the second sampling round less often agreed that the risk of getting infected with COVID-19 was part of their job, compared to HCWs from the first sampling round (69% versus 84%, respectively).

**Sense of control in getting infected.** Overall, 40% (869/2199) of HCWs indicated that they felt that that getting infected with COVID-19 was outside of their control. This feeling was more prevalent in junior than in senior HCWs (46% versus 38%, *P* value < .01), and was significantly associated with a WHO-5 score below 50 points (aOR 2.1, 95% CI 1.7–2.6). In

**Table 1. Demographic information of all participating hospital health care workers (HCWs).**

|  | Hospital HCWs N = 2289 (%)[a] |
|---|---|
| **Age, mean (±SD)** | 42 (11) |
| **Female** | 1509 (66) |
| **Region**[1] |  |
| Southern Europe | 1244 (54) |
| Western Europe | 658 (29) |
| Northern Europe | 329 (14) |
| Eastern Europe | 57 (3) |
| **Living with others** | 1916 (85) |
| **Informal care responsibilities for any other adults** | 614 (27) |
| **Academic hospital** | 1570 (70) |
| **Medical specialty** |  |
| Acute care (anaesthesiology, ER, ICU) | 748 (33) |
| Internal medicine | 506 (22) |
| Surgery | 226 (10) |
| Paediatrics | 144 (6) |
| Other | 665 (29) |
| **Job role** |  |
| Junior nurse | 240 (11) |
| Senior nurse | 657 (29) |
| Senior medical doctor | 803 (35) |
| Junior medical doctor | 269 (12) |
| Junior allied health professional | 31 (1) |
| Senior allied health professional | 66 (3) |
| Other | 223 (10) |
| **Daily patient contact** | 1844 (81) |
| **Direct COVID-19 patient care** | 1699 (74) |

COVID-19, coronavirus diseases 2019; ER, emergency room; HCW, health care worker, ICU, intensive care unit; IQR, interquartile range; SD, standard deviation.

[a] Sub division of Europe adapted from the United Nations; for the current study, Cyprus, Israel and Turkey were categorized as Southern Europe [25].

multivariable regression, factors that were positively associated with a perceived sense of control over getting infected with COVID-19 were having received sufficient COVID-19 IPC training, having greater confidence in using PPE and perceived institutional trust (Table 3).

## Behavioural influences on following IPC guidance

The majority of HCWs had received training related to general IPC procedures and indicated that there were clear policies and protocols for everyone to follow related to COVID-19 IPC procedures, namely 75% (n = 1725) and 80% (n = 1785), respectively (Fig 1). Twenty-three percent (n = 391) of HCWs that cared for COVID-19 patients indicated they had not received sufficient training in IPC practices specific to COVID-19.

Most HCWs (1814, 79%) felt that following IPC recommendations would protect them from becoming ill with COVID-19, and almost all (2134, 96%) intended to always use recommended PPE when taking care of patients with suspected or confirmed COVID-19, when having access to these. HCWs reported positive social influences at work, such as colleagues regularly following IPC measures and encouragement by senior medical/nurse staff to follow recommended procedures (Fig 1). Trust in their health facility to be competent, honest with staff, and act in the best interest of its staff when managing the response to COVID-19 differed per region, being 61%, 67%, 79%, 81% for Eastern, Southern, Northern and Western European HCWs, respectively.

**Table 2. WHO-5 emotional Well-Being Index per subgroup.**

| | Mean (±SD) | P value |
|---|---|---|
| **Gender** | | |
| Male | 59.6 (19.3) | < .001 |
| Female | 54.5 (19.0) | |
| **Job role** | | |
| Junior nurse | 54.0 (18.2) | .04 |
| Senior nurse | 57.5 (19.3) | |
| Junior medical doctor | 54.8 (17.8) | |
| Senior medical doctor | 56.9 (19.8) | |
| Junior allied health professional | 57.5 (17.2) | |
| Senior allied health professional | 51.0 (21.2) | |
| Other | 55.7 (19.8) | |
| **Region[a]** | | |
| Eastern Europe | 52.7 (19.9) | < .001 |
| Southern Europe | 53.4 (19.5) | |
| Northern Europe | 56.0 (18.8) | |
| Western Europe | 62.1 (17.8) | |
| **COVID-19 direct patient care** | | |
| HCWs with COVID-19 patient care | 56.0 (19.5) | NS |
| HCWs without COVID-19 patient care | 57.0 (18.7) | |

COVID-19, coronavirus disease 2019; HCW, health care worker; NS, not significant; SD, standard deviation; WHO, World Health Organization.

[a] Sub division of Europe adapted from the United Nations; for the current study, Cyprus, Israel and Turkey were categorized as Southern Europe [25].

**Use and availability of personal protective equipment.** Based on self-reported use of PPE at last contact with a patient suspected or confirmed with COVID-19, HCWs reported good compliance with PPE recommendations as provided by the WHO (Fig 2). A larger

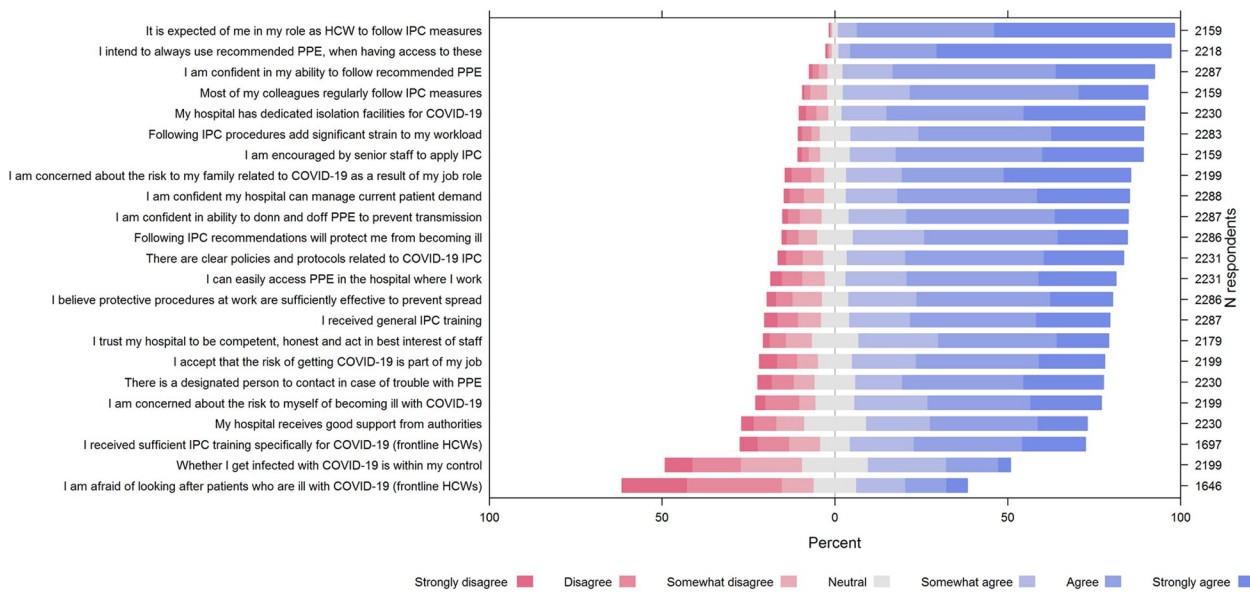

**Fig 1. Perceptions of hospital healthcare workers on recommended IPC procedures, perceived skills, intentions and environmental resources.** HCWs, health care workers; IPC, infection prevention and control; PPE; personal protective equipment. NB. Individual statements were abbreviated for readability of this figure (see S1 File for all complete statements used in the surveys).

**Table 3. Ordered logistic regression for the effect of perceived skills, self-reported environmental context, social influences and institutional trust on a positive sense of control over getting infected with COVID-19[a].**

|  | aOR | 95% CI | | P value |
|---|---|---|---|---|
| Having received general training for IPC procedures for communicable diseases | 0.98 | 0.90 | 1.06 | NS |
| Having received sufficient training in IPC practices for COVID-19 | 1.11 | 1.02 | 1.20 | .01 |
| Feeling confident in ability to correctly use PPE | 1.08 | 1.00 | 1.17 | .045 |
| Index of PPE availability | 1.05 | 0.96 | 1.15 | NS |
| Feeling encouraged and supported by senior medical/nurse staff to apply recommended IPC measures | 1.03 | 0.96 | 1.11 | NS |
| Trust that health facility is competent, honest and acts in best interest of its staff | 1.34 | 1.24 | 1.45 | < .0001 |

aOR, adjusted odds ratio; COVID-19, coronavirus disease 2019; CI, confidence interval; IPC, infection prevention and control; PPE, personal protective equipment; NS, not significant.

[a] This model was adjusted for age, gender, living situation, European region and providing direct care for COVID-19 patients.

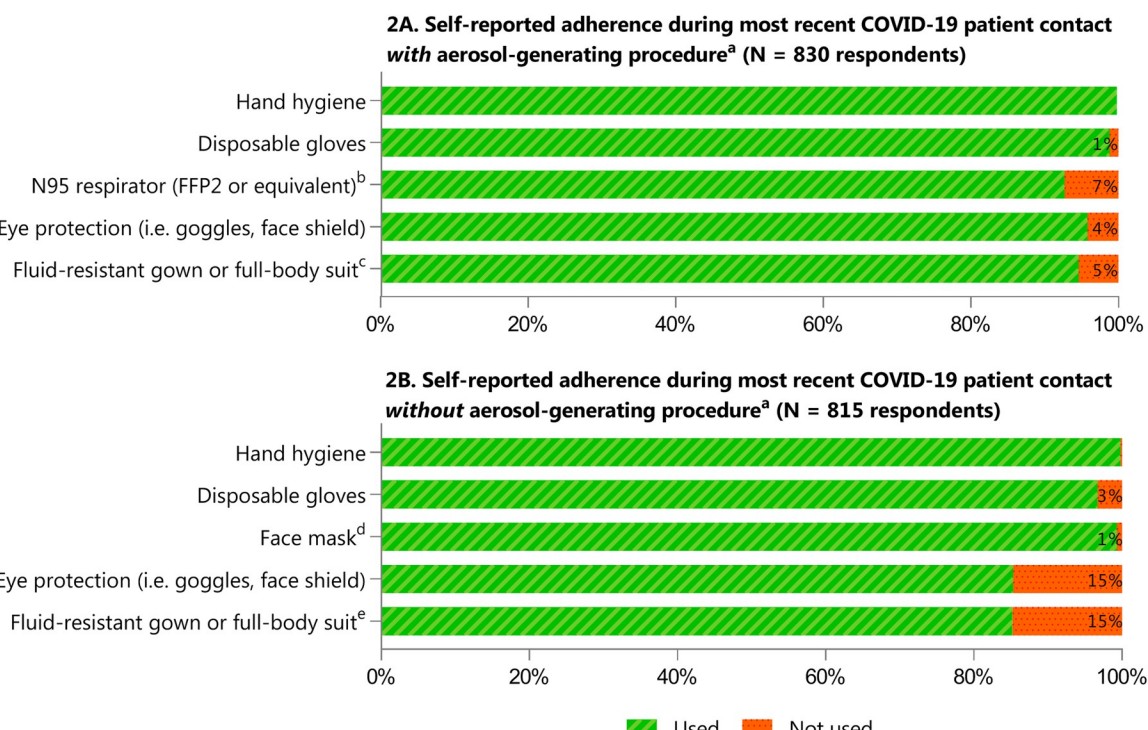

**Fig 2. Self-reported adherence to recommended IPC procedures[a] used during most recent clinical contact with COVID-19 case.** [a] Recommended infection prevention and control (IPC) procedures during direct medical care with suspected or confirmed COVID-19 patients according to WHO interim guidance document (6 April 2020) [26] on the rational use of personal protective equipment (PPE) at the time of data collection: (A) contacts with aerosol-generating procedures (e.g. tracheal intubation, non-invasive ventilation, cardiopulmonary resuscitation): Gloves, N95 mask or equivalent, eye protection (i.e. goggles or face shield), and fluid-resistant long-sleeved gown. (B) Contacts without aerosol-generating procedure: Gown, gloves, medical mask and eye protection (goggles or face shield). Hand hygiene was part of recommendations during all contacts. [b] Of those respondents (n = 60) that did not use an N95 respirator, 58 (97%) used another type of face mask, such as surgical mask (categorized in this figure as 'no'). [c] Of those respondents (n = 44) that did not use either a fluid-resistant gown or full-body suit, 26 (59%) used a disposable plastic apron (categorized in this figure as 'no'). [d] Of those respondents use either a gown or full-body suit (n = 120), 68 (57%) used a disposable plastic apron. [e] Of those HCWs that used a face mask (n = 802), 69% used an N95 mask (n = 557) and 31% (n = 245) used another type of mask, such as a surgical mask.

proportion of HCWs reported limited access to PPE materials at work during the first sampling round compared to the second survey round, namely 28% (52/183) versus 14% (293/2048), respectively. Overall, PPE that was most often reported to be in limited or no supply during the HCWs' most recent clinical shift were N95 respirators (30%), fluid-resistant gowns (25%) and eye protection (i.e. goggles) (21%) (S1 Fig).

## Discussion

In this study, we found that European hospital HCWs reported low levels of emotional wellbeing while providing hospital care and following local IPC recommendations during the start of the COVID-19 pandemic, with particular concerns for those providing direct COVID-19 patient care, junior staff and women.

There is more evidence emerging on the psychological strain on HCWs working during the first wave of the COVID-19 pandemic [3,7,27]. A systematic review and meta-analysis investigating the psychological impact during the first months of COVID-19 predominantly in Asia, identified high levels of depression among medical staff [3]. Across Europe, we similarly identified psychological strain in hospital HCWs. Not only were there high levels of concern about HCWs' own risk of COVID-19 infection, almost all respondents were worried about the risk to their families due to their job role. One-third of HCWs reported fear of looking after COVID-19 patients. Further, when quantifying emotional wellbeing using the WHO-5 Well-Being Index, we found that more than one-third of HCWs had a score below 50 points, indicative of depression symptoms [23]. These findings reiterate worrisome findings of other studies performed during the start of the COVID-19 escalation in Europe [28–31].

It is known that health emergencies can magnify already existing structural inequities [10]. Our study highlights impacts for women in particular, who comprise the majority of all HCWs worldwide. In general, women perform the majority of care within the home, are more likely to care for family members ill with COVID-19, and may face gender bias at work [10,32]. In our study, female HCWs reported higher levels of emotional strain compared to males, echoing findings from other COVID-19 studies [9,33–36]. These results highlight the need for awareness of the differential psychological impacts on HCWs during the COVID-19 pandemic and future pandemics.

IPC recommendations aim to protect patients, HCWs, and the health system, and are therefore tied to risk perception as recommendations should ensure that HCWs feel protected during their work. Perceived lack of control is known to be an important stressor of mental health and is associated with occupational burnout [37–39]. Research during the 2002–2003 SARS epidemic suggested that the combined role of three specific institutional measures impacted HCWs' risk perception (including avoidance of patients and acceptance of risk) and belief in the effectiveness of protective measures at work: clear policies and protocols, available specialists, and adequate training [40]. In our study, junior HCWs in particular felt that getting infected with COVID-19 was outside of their control. Risk perception is complex and multifactorial. A qualitative study among frontline HCWs who chose to respond to the West Africa Ebola virus disease outbreak in 2014–2016, showed that next to individual and social-level factors, institutional trust was a key risk attenuator [41]. Our multivariable analysis also identified organisational and personal factors that were positively associated with feelings of having control over becoming infected, including COVID-19 IPC training, confidence in the use of PPE as well as institutional trust. These aspects may therefore be important to consider in IPC implementation.

Overall, our survey suggests that institutions should adopt a multifaceted approach in IPC preparedness and training in order to best support hospital HCWs at work during an

infectious diseases pandemic. Such an approach begins with recognition of the importance of preventing and mitigating adverse impacts on wellbeing, together with the explicit assurance to protect the clinical workforce. These foundations for protecting physical and mental health protection are made through ensuring safe working conditions and accessible support. We echo recommendations that local IPC guidelines should be clearly communicated, and HCWs should receive specific IPC training, have access to appropriate PPE, and feel confident in its use [42]. Crucially, we add that health facilities must be aware of, and address differential impacts and experiences, such as those of female and junior HCWs. Given the already-heavy burden and time pressures on HCWs, effective support must therefore be readily accessible and responsive to the different needs, pressures, and barriers experienced by different groups. Together, these measures should work to increase HCWs' feelings of self-efficacy to undertake IPC behaviors and, subsequently, their perceived safety.

Previous studies have also shown the importance of managers' support and a safe workplace culture [42]. To identify where most improvements could be made, we further suggest that health facilities rapidly assess local perceptions of their HCWs on IPC recommendations and their wellbeing using the WHO research template and data collection tool (freely available online) [16]. This would serve to identify specific key local needs that are/are not being met, enabling wellbeing support services to be better targeted and tailored, as research has shown that these are often poorly utilised by HCWs [9].

There are important limitations of the current study that should be acknowledged. First, in the first sampling round we chose to rapidly recruit HCWs from existing research networks where we could accurately describe our study base population, limiting the number of HCWs we could invite for participation. Along with data collection taking place during the first peak of the pandemic in Europe and restricting to a single reminder because of high clinical workload of HCWs, this ultimately led to small sample size. Therefore, a second sampling round was performed using convenience sampling, leading to two different sampling methods used in the current study. Notably, the target population remained the same. Second, inherent to our survey study design there is an important risk of response bias, both on the level of demographic characteristics as well as on personal views, meaning that our responding HCWs do not reflect an representative sample of the entire European hospital HCW population. The choice of survey methodology over qualitative design was, in part, driven by a need to rapidly collect data that could be used to inform operational pandemic response. Survey methods will miss more nuanced accounts that could help explain our findings which would be best captured through qualitative methods. A free text box allying respondents to add their reflections would have been helpful to enable us to capture some of these reflections. Third, this was a cross-sectional observational study using quantitative data, we therefore cannot draw strong conclusions on the direction of effects. Lastly, we were unable to triangulate responses with actual local PPE supplies and hospital infection rates.

## Conclusions

In conclusion, this international cross-sectional survey examined the perceptions of European hospital HCWs on their emotional wellbeing and local recommended IPC procedures during the onset of the COVID-19 pandemic. We advocate that hospitals should provide multifaceted IPC training that accounts for behavioural determinants and tailored support that accounts for the needs of junior and female HCWs. Such training should be inclusive and accessible to all. Further, local hospital support systems should be intensified and health facilities must be aware of, address, and provision for potential differential impacts to better care for our HCWs during the current COVID-19 and potential future pandemics.

## Supporting information

**S1 Fig. Self-reported availability of personal protective equipment during most recent clinical shift.**
(DOCX)

**S1 Table. Country of work of responding hospital healthcare workers.**
(DOCX)

**S2 Table. Response bias assessment first sampling round.**
(DOCX)

**S3 Table. Multivariable linear regression for the effect of gender on WHO-5 emotional wellbeing index of hospital healthcare workers.**
(DOCX)

**S1 File. Survey tools used in the current study.**
(DOCX)

**S2 File. Principal component analysis.**
(DOCX)

## Acknowledgments

We would sincerely like to thank Cristina Prat (UMC Utrecht, Netherlands), Javier Muñoz Bravo (CIBER, www.ciberes.org) and Branka Nikolić (Clinic Narodni Front, Serbia) for their important contributions in the distribution of this survey. We would also like to thank Jelle Lyskawa (UMC Utrecht, Netherlands) and Melanie Hoste (University of Antwerp, Belgium) for kindly assisting in the execution of this study. We acknowledge Alice Simniceanu, April Baller, Benedetta Allegranzi, Maria Clara Padoveze, Yolanda Bayugo (World Health Organization) for key contributions to the development of the WHO study protocol. Lastly, we would like to express our gratitude to all responding health care workers who took the time to participate in our study, despite the busy times at the outlook of the COVID-19 pandemic.

## Author Contributions

**Conceptualization:** Denise van Hout, Paul Hutchinson, Marta Wanat, Herman Goossens, Sibyl Anthierens, Sarah Tonkin-Crine, Nina Gobat.

**Data curation:** Denise van Hout, Nina Gobat.

**Formal analysis:** Denise van Hout, Paul Hutchinson, Marta Wanat, Caitlin Pilbeam, Sibyl Anthierens, Sarah Tonkin-Crine, Nina Gobat.

**Funding acquisition:** Herman Goossens, Sibyl Anthierens, Nina Gobat.

**Investigation:** Denise van Hout, Paul Hutchinson, Marta Wanat, Caitlin Pilbeam, Sibyl Anthierens, Sarah Tonkin-Crine, Nina Gobat.

**Methodology:** Denise van Hout, Paul Hutchinson, Marta Wanat, Caitlin Pilbeam, Sibyl Anthierens, Sarah Tonkin-Crine, Nina Gobat.

**Project administration:** Denise van Hout.

**Resources:** Herman Goossens.

**Software:** Paul Hutchinson.

**Supervision:** Sibyl Anthierens, Sarah Tonkin-Crine, Nina Gobat.

**Validation:** Paul Hutchinson, Marta Wanat, Caitlin Pilbeam, Sibyl Anthierens, Sarah Tonkin-Crine, Nina Gobat.

**Writing – original draft:** Denise van Hout.

**Writing – review & editing:** Denise van Hout, Paul Hutchinson, Marta Wanat, Caitlin Pilbeam, Herman Goossens, Sibyl Anthierens, Sarah Tonkin-Crine, Nina Gobat.

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
