## [Decision Letter · Decision Letter 0]

8 Sep 2021

PONE-D-20-40846

The experience of European hospital-based health care workers on following infection prevention and control procedures for COVID-19

PLOS ONE

Dear Dr. van Hout,

Thank you for submitting your manuscript to PLOS ONE. After careful consideration, we feel that it has merit but does not fully meet PLOS ONE’s publication criteria as it currently stands. Therefore, we invite you to submit a revised version of the manuscript that addresses the points raised during the review process.

We look forward to receiving your revised manuscript.

Kind regards,

Amitava Mukherjee, ME, Ph.D.

Academic Editor

PLOS ONE

Journal Requirements:

Reviewers' comments:

Reviewer's Responses to Questions

**Comments to the Author**

1. Is the manuscript technically sound, and do the data support the conclusions?

Reviewer #1: Partly

2. Has the statistical analysis been performed appropriately and rigorously? 

Reviewer #1: I Don't Know

3. Have the authors made all data underlying the findings in their manuscript fully available?

Reviewer #1: Yes

4. Is the manuscript presented in an intelligible fashion and written in standard English?

Reviewer #1: Yes

5. Review Comments to the Author

Reviewer #1: The study benefitted from a good sample size and from addressing topical and relevant issues (e.g., healthcare staff wellbeing and infection and prevention controls). However, I felt confused throughout as to what the main goals of the study were – it felt like a mishmash of questions had been asked and statistics had been conducted. I also felt concerned as to the factor analysis which is briefly referenced and minimally described but which is of key importance to the overall study (as it was the basis for developing the main questionnaire used in the study). In general, it seemed a lot of survey questions had been asked but not enough thought had been put into making the data into a clear and cohesive study. Specific comments below:

Abstract:

1. The aims are not clear- what were the specific purposes of doing this survey? What did the authors want to learn? Were there any predictions?

2. The reason for having two ‘rounds’ is also unclear. As all the results seem to only refer to one group, were the two ‘rounds’ simply two different ways of recruiting participants and the two groups were then pooled for the purposes of the analysis?

Intro

The intro covers some relevant issues – for example, demonstrating that HCWs are at heightened risk of both catching Covid and experiencing burnout. It stresses that HCWs should be familiar with IPC procedures, and that women may be at higher risk of experiencing stress and burnout. However, there are a couple of issues with the introduction:

3. It is unclear what the gaps in knowledge are as the introduction progresses. The authors need to more clearly demonstrate what IS NOT known which the present research aimed to address

4. The aims are poorly written. They are extremely broad and do not give the reader a sense of where the analyses and results will be heading. It would be acceptable to have broad aims but to follow these up with specific objectives.

5. The use of a survey methodology also needs to be justified – both of the current aims strike me as being better suited to a qualitative study. Again, the gap in the literature that this study (using a survey approach) addresses needs to be clear

Methods

6. The section on the survey tool is confusing. It says the tool was developed ‘to capture theoretically informed influences on HCWs’ motivation, opportunity and ability to follow general IPC precautions’. There needs to be info on the number of items, how many captured each of these three things (motivation, opportunity and ability), and some example questions. There also needs to be basic info about the psychometrics of the survey once it had been developed for use in round 2 of the research (i.e., following the first round of the research). It is not sufficient to say the authors didn’t have this information to begin with.

7. The section on the survey tool is also confusing because at the end of the paragraph the authors throw in that the purpose of the first round of the survey was to develop and refine the tool which is used in round 2. This should have been clear from the abstract and the introduction. If such a tool (which is the foundation of the research) didn’t exist prior to the survey being conducted, then one of the aims of the study was surely to develop such a measure? Aside from this only one other questionnaire is used, so it is important that this is clarified.

8. The statistical analyses section refers to objectives – but these objectives are not outlined anywhere earlier in the paper. These objectives need to be provided at the end of the introduction. The methods is not the place to introduce new objectives.

9. ‘bivariate analyses for gender, job role, European region and providing COVID-19 care by

independent sample T-test or one-way ANOVA’: when were independent sample T-tests used and when were ANOVAS used? This needs to be clarified

10. The statistical analyses section also describes conducting tests relating to examining gender etc differences in wellbeing and factors relating to ‘HCWs’ perceived sense of control over getting infected with COVID-19’. Due to unclear aims and objectives earlier in the paper, these analyses seem to ‘come out of nowhere’. Really the issue here is not what is in the statistical analyses section but the foregoing information in the paper. There is a lack of build-up and explanation in the intro to prepare readers for these analyses and they are not clear in the aims/objectives of the study.

11. Halfway through a paragraph in the statistical analyses section, a factor analysis is mentioned with minimal information. If the purpose of the factor analysis is to develop the main tool which is used in the paper (which it is) I would expect much, much more information about this – the type of analysis done, the scree plot, the factor loadings etc. As it is, we don’t know from the manuscript even how many items the authors started with and the approach they used to reducing the number of items they had. This is very concerning. I would expect to see all the original items in a table with their factor loadings and then the approach used for reducing these. I’d also expect to see some kind of justification for the sample size.

12. Information about how much data was missing and how this was managed needs to be provided in the statistical analyses section

6. PLOS authors have the option to publish the peer review history of their article (what does this mean?). If published, this will include your full peer review and any attached files.

Reviewer #1: No

---

## [Author Response · Author response to Decision Letter 0]

17 Oct 2021

Reviewer #1: The study benefitted from a good sample size and from addressing topical and relevant issues (e.g., healthcare staff wellbeing and infection and prevention controls). However, I felt confused throughout as to what the main goals of the study were – it felt like a mishmash of questions had been asked and statistics had been conducted. I also felt concerned as to the factor analysis which is briefly referenced and minimally described but which is of key importance to the overall study (as it was the basis for developing the main questionnaire used in the study). In general, it seemed a lot of survey questions had been asked but not enough thought had been put into making the data into a clear and cohesive study. Specific comments below:

Author’s response: We would sincerely like to thank the reviewer for their valuable time and we sincerely appreciate the careful reading of our manuscript. In the following we will address all comments point by point.

Abstract:

1. The aims are not clear- what were the specific purposes of doing this survey? What did the authors want to learn? Were there any predictions?

Author’s response: The study took place at a crucial time in the peak of the first wave of the pandemic in order to rapidly respond on an international level to the needs of the healthcare workers. The primary aim of the study was to capture health worker perceptions related to key factors that are known to influence their adherence to infection prevention and control guidelines. The study was rapidly developed at the start of the pandemic, and, in response to emerging evidence of the impact of the pandemic on health worker mental health and emotional well-being, a second objective was included to capture data related to wellbeing. 

Changes to the manuscript: To clarify these objectives in the abstract, we have updated the text – please see p. 2, lines 28-31 of the revised manuscript.

2. The reason for having two ‘rounds’ is also unclear. As all the results seem to only refer to one group, were the two ‘rounds’ simply two different ways of recruiting participants and the two groups were then pooled for the purposes of the analysis?

Authors’ response: This was a rapidly developed study, designed to respond quickly to emerging needs at the start of the pandemic. The first round was conducted during the first peak pandemic wave in Europe. We had a low response rate to this survey at round 1 – in part due to competing clinical pressures during this time. For the second round of data collection, we adapted our approach to enhance recruitment.. This second round of data collection was undertaken toward the end of the first pandemic wave in Europe, and we had a strong response. Importantly, the target population of both rounds were the same, only our recruitment approach changed.

Changes to the manuscript: To clarify this further, we have updated the abstract – please see p. 2, lines 32-35 and we have added information to the methods section – please see p. 6, lines 113-117.

Intro

The intro covers some relevant issues – for example, demonstrating that HCWs are at heightened risk of both catching Covid and experiencing burnout. It stresses that HCWs should be familiar with IPC procedures, and that women may be at higher risk of experiencing stress and burnout. However, there are a couple of issues with the introduction:

3. It is unclear what the gaps in knowledge are as the introduction progresses. The authors need to more clearly demonstrate what IS NOT known which the present research aimed to address

Authors’ response: We thank the reviewer for this comment. We have added more specific statements about the research gaps that our study aimed to fulfill in to the Introduction section of the paper. 

Changes to the manuscript: please see p. 4, lines 68-69 and p. 5, lines 89-93 of the track changes version of the manuscript.

4. The aims are poorly written. They are extremely broad and do not give the reader a sense of where the analyses and results will be heading. It would be acceptable to have broad aims but to follow these up with specific objectives.

Authors’ response: We thank the reviewer for this suggestion. We acknowledge the aims of our study were indeed broad and mostly descriptive of nature. We also agree that our specific objective to look at subgroups, such as potential differences in perceptions of male and female HCWs, were not clearly described to readers in the Introduction section. We have therefore updated the Introduction of the manuscript accordingly. 

Changes to the manuscript: see p. 5, lines 97-101 of the track changes version of the manuscript.

5. The use of a survey methodology also needs to be justified – both of the current aims strike me as being better suited to a qualitative study. Again, the gap in the literature that this study (using a survey approach) addresses needs to be clear.

Authors’ response: Thank you for this remark. The choice of survey methodology was driven, in part, by a pragmatic need to rapidly collect data that could be used to inform operational response. The dimensions included in the survey linked with key factors that are known to influence adherence to infection control guidelines and which can then guide specific actions to improve staff adherence to guidelines.We agree that a qualitative study related to these issues would have provided rich insights into many of the issues raised. Indeed, many of the other groups that implemented this survey in other parts of the world as part of the WHO R&D Blueprint ‘COVID-19 Social Science in Outbreaks’ (the study protocol and survey tool are freely available), included a free text open comments box in their surveys and elicited interesting findings. However, in our case, by using a survey design we were able to acquire a large sample size and international reach, which would have been far smaller if we would have used a qualitative study. It is also important to note that during our study period, at the beginning of the COVID-19 pandemic in Europe, there was already a high burden on healthcare professionals. Survey methods are much lower commitment for participants already under strain. 

Changes to the manuscript: we have added the rationale for a survey study design to the Methods section, see p. 15, lines 333-335 of the track changes version of the manuscript. 

Methods

6. The section on the survey tool is confusing. It says the tool was developed ‘to capture theoretically informed influences on HCWs’ motivation, opportunity and ability to follow general IPC precautions’. There needs to be info on the number of items, how many captured each of these three things (motivation, opportunity and ability), and some example questions. There also needs to be basic info about the psychometrics of the survey once it had been developed for use in round 2 of the research (i.e., following the first round of the research). It is not sufficient to say the authors didn’t have this information to begin with.

Authors’ response: This study was rapidly conceived and developed at the start of the COVID-19 pandemic to respond to a need to better understand how prepared health workers were to adhere to infection prevention and control guidelines. At the point of protocol development, a well-developed and pre-validated tool to capture behavioral determinants of guideline adherence was not in existence. We therefore rapidly developed a tool based on expert opinion and guided by the Theoretical Domains Framework. Data from the first round of data collection in this study were used to explore the psychometric properties of the tool, not only for the second round of sampling but also for groups implementing the same survey in different parts of the world. 

Changes to the manuscript: no changes were made to the manuscript, the information above is already depicted in the original paper, please see p. 6-7, lines 124-135.

7. The section on the survey tool is also confusing because at the end of the paragraph the authors throw in that the purpose of the first round of the survey was to develop and refine the tool which is used in round 2. This should have been clear from the abstract and the introduction. If such a tool (which is the foundation of the research) didn’t exist prior to the survey being conducted, then one of the aims of the study was surely to develop such a measure? Aside from this only one other questionnaire is used, so it is important that this is clarified.

Authors’ response: We apologize for any confusion. The purpose of the first round of the survey was not specifically “to develop” the tool for a second round of sampling within our study (ie. the second round with different sampling method was performed because of low response rate). However, when deciding on a second round of sampling we did use data from the first round of data collection to explore the psychometric properties of the tool and to refine the items for data collection, also for groups implementing the same survey in different parts of the world. See also the explanation above. We hope this clarifies.

8. The statistical analyses section refers to objectives – but these objectives are not outlined anywhere earlier in the paper. These objectives need to be provided at the end of the introduction. The methods is not the place to introduce new objectives.

Authors’ response: Thank you for this comment. The respective statistical analyses section refers to the objectives stated at the end of the Introduction section of the paper, which were previously there referred to as ‘aims’. We acknowledge this was not clear to readers.

Changes to the manuscript: we have updated the text and clarified the objectives – please see p. 5, lines 97-101 of the track changes version of the manuscript).

9. ‘bivariate analyses for gender, job role, European region and providing COVID-19 care by

independent sample T-test or one-way ANOVA’: when were independent sample T-tests used and when were ANOVAS used? This needs to be clarified

Authors’ response: an independent sample T-test is carried out to investigate two groups, whereas ANOVA is used to compare multiple (>2) groups (ANOVA is equivalent to running multiple T-tests). We have clarified this in our manuscript. 

Changes to the manuscript: please see p. 7, lines 162-163 of the track changes version of the manuscript).

10. The statistical analyses section also describes conducting tests relating to examining gender etc differences in wellbeing and factors relating to ‘HCWs’ perceived sense of control over getting infected with COVID-19’. Due to unclear aims and objectives earlier in the paper, these analyses seem to ‘come out of nowhere’. Really the issue here is not what is in the statistical analyses section but the foregoing information in the paper. There is a lack of build-up and explanation in the intro to prepare readers for these analyses and they are not clear in the aims/objectives of the study.

Authors’ response: We agree with the reviewer that these analyses should have been more clearly introduced in the Introduction section of the paper.

Changes to the manuscript: please see p. 4, lines 68-70 and p. 5, 97-101 of the track changes version of the manuscript.

11. Halfway through a paragraph in the statistical analyses section, a factor analysis is mentioned with minimal information. If the purpose of the factor analysis is to develop the main tool which is used in the paper (which it is) I would expect much, much more information about this – the type of analysis done, the scree plot, the factor loadings etc. As it is, we don’t know from the manuscript even how many items the authors started with and the approach they used to reducing the number of items they had. This is very concerning. I would expect to see all the original items in a table with their factor loadings and then the approach used for reducing these. I’d also expect to see some kind of justification for the sample size. 

Authors’ response: We apologize for any confusion. In contrary to the reviewer’s comment, the factor analysis was not performed “to develop” the main tool, but rather to evaluate the psychometric properties of the tool after the first round of data collection was performed. The data collection tool was developed rapidly based on literature review and expert opinion (as explained above and in p. 6-7, lines 124-135 of the Methods section). As mentioned in the paper, no formal sample size was performed before the start of this study (see p. 9, lines 185-186).

Changes to the manuscript: We have added the following to the paragraph to provide further explanation for the use of principal component factor analysis, which was used to develop variables measuring beliefs about PPE effectiveness, PPE availability, skills, and institutional trust in the multivariable models (see p. 8-9, lines 172-184). 

From the Likert-scale questions, we conducted principal component factor analysis to examine correlation matrices and construct indices from the first principal component for the following constructs: beliefs about effectiveness of PPE, availability of PPE at the respondent’s institution, skills, and institutional trust test. The first principal component for each of these constructs was then included as an explanatory variable in the multivariable analysis. This allowed us to test, for example, for the independent effect of institutional trust, on the dependent variable, HCWs’ perceived sense of control over getting infected with COVID-19, while control for other regression model covariates. For each of the constructed indices, Cronbach’s alpha scores for the Likert-scale components exceeded 0.75. 

12. Information about how much data was missing and how this was managed needs to be provided in the statistical analyses section

Authors’ response: we thank the reviewer for this important comment. As mentioned in the Methods section, surveys with completion of only the first part of the survey containing demographic and basic information about IPC training (corresponding to survey completion <58%) were excluded from data analyses. Due to mid- or end-survey dropout, the denominator per statement is different. Therefore, the total number of respondents per statement is provided in Fig 1, and if otherwise, in the manuscript body. For example in case of perceptions about emotional wellbeing, the sentence “There were 2180 (95%) HCWs who completed all questions about emotional wellbeing” is provided. In case of other denominators (for example when analyzing subgroups), this is also mentioned specifically in the manuscript text. We agree with the reviewer that this information should also be available in the Methods section.

Changes to the manuscript: please see p. 9, lines 188-190 of the track changes version of the manuscript).

---

## [Decision Letter · Decision Letter 1]

17 Dec 2021

PONE-D-20-40846R1

The experience of European hospital-based health care workers on following infection prevention and control procedures and their wellbeing during the first wave of the COVID-19 pandemic

PLOS ONE

Dear Dr. van Hout,

Thank you for submitting your manuscript to PLOS ONE. After careful consideration, we feel that it has merit but does not fully meet PLOS ONE’s publication criteria as it currently stands. Therefore, we invite you to submit a revised version of the manuscript that addresses the points raised during the review process.

We look forward to receiving your revised manuscript.

Kind regards,

Amitava Mukherjee, ME, Ph.D.

Academic Editor

PLOS ONE

Journal Requirements:

Reviewers' comments:

Reviewer's Responses to Questions

**Comments to the Author**

1. If the authors have adequately addressed your comments raised in a previous round of review and you feel that this manuscript is now acceptable for publication, you may indicate that here to bypass the “Comments to the Author” section, enter your conflict of interest statement in the “Confidential to Editor” section, and submit your "Accept" recommendation.

Reviewer #2: (No Response)

2. Is the manuscript technically sound, and do the data support the conclusions?

Reviewer #2: Yes

3. Has the statistical analysis been performed appropriately and rigorously? 

Reviewer #2: Yes

4. Have the authors made all data underlying the findings in their manuscript fully available?

Reviewer #2: Yes

5. Is the manuscript presented in an intelligible fashion and written in standard English?

Reviewer #2: Yes

6. Review Comments to the Author

Reviewer #2: This manuscript assesses the perception of hospital HCWs on the adequacy of local IPC measures and concludes that institutions should adopt a multifaceted approach in IPC preparedness and training in order to best support hospital HCWs at work during an infectious diseases pandemic. The authors discuss factors that contribute to staff wellbeing and the perceived efficacy of PPE and the results are clearly presented.

Minor comments: While a PCA is mentioned in the methods section, there are no details on the results of the PCA and a more detailed description of this analysis should be added as a supplementary file.

The background section of the abstract would be strengthened if the rational behind the study was discussed in more detail.

7. PLOS authors have the option to publish the peer review history of their article (what does this mean?). If published, this will include your full peer review and any attached files.

Reviewer #2: No

---

## [Author Response · Author response to Decision Letter 1]

10 Jan 2022

Reviewer #2: This manuscript assesses the perception of hospital HCWs on the adequacy of local IPC measures and concludes that institutions should adopt a multifaceted approach in IPC preparedness and training in order to best support hospital HCWs at work during an infectious diseases pandemic. The authors discuss factors that contribute to staff wellbeing and the perceived efficacy of PPE and the results are clearly presented.

Author’s response: We would sincerely like to thank the reviewer for their valuable time and we sincerely appreciate the careful reading of our manuscript. In the following we will address all comments point by point.

Minor comments: While a PCA is mentioned in the methods section, there are no details on the results of the PCA and a more detailed description of this analysis should be added as a supplementary file.

Author’s response: we thank the reviewer for this comment. The principal component factor analysis was performed to construct indices from the first principal component for the following measures: beliefs about effectiveness of PPE, availability of PPE at the respondent’s institution, and skills for preparedness for dealing with COVID-19. The index constructed from the first principal component for each of these measures was then included as an explanatory variable in the multivariable analysis. We agree with the reviewer that details on data-analysis and results of the PCA were missing in the manuscript.

Changes to the manuscript: details on the PCA were clarified in the main paper (please see p. 8, lines 168-174), and a detailed description of the PCA was added as a new supplementary file – please see “S2 File”.

The background section of the abstract would be strengthened if the rational behind the study was discussed in more detail.

Changes to the manuscript: To clarify the rationale of the study in the abstract, we have updated the text – please see p. 2, lines 29-31 of the revised manuscript.

---

## [Editor Report · Decision Letter 2]

17 Jan 2022

The experience of European hospital-based health care workers on following infection prevention and control procedures and their wellbeing during the first wave of the COVID-19 pandemic

PONE-D-20-40846R2

Dear Dr. van Hout,

We’re pleased to inform you that your manuscript has been judged scientifically suitable for publication and will be formally accepted for publication once it meets all outstanding technical requirements.

Kind regards,

Amitava Mukherjee, ME, Ph.D.

Academic Editor

PLOS ONE
---

## [Editor Report · Acceptance letter]

27 Jan 2022

PONE-D-20-40846R2 

The experience of European hospital-based health care workers on following infection prevention and control procedures and their wellbeing during the first wave of the COVID-19 pandemic 

Dear Dr. van Hout:

I'm pleased to inform you that your manuscript has been deemed suitable for publication in PLOS ONE. Congratulations! Your manuscript is now with our production department. 

Kind regards, 

on behalf of

Professor Dr. Amitava Mukherjee 

Academic Editor

PLOS ONE